# Impact of Climate Change on Agricultural Droughts in Spain

**María del Pilar Jiménez-Donaire [1,\*], Juan Vicente Giráldez [1,2] and Tom Vanwalleghem [1]**

[1] Department of Agronomy, University of Córdoba, 14071 Córdoba, Spain; ag1gicej@uco.es (J.V.G.); ag2vavat@uco.es (T.V.)

[2] Institute for Sustainable Agriculture, CSIC, 14071 Córdoba, Spain

\* Correspondence: g52jidom@uco.es

**Abstract:** Drought is an important natural hazard that is expected to increase in frequency and intensity as a consequence of climate change. This study aimed to evaluate the impact of future changes in the temperature and precipitation regime of Spain on agricultural droughts, using novel static and dynamic drought indices. Statistically downscaled climate change scenarios from the model HadGEM2-CC, under the scenario representative concentration pathway 8.5 (RCP8.5), were used at a total of 374 sites for the period 2006 to 2100. The evolution of static and dynamic drought stress indices over time show clearly how drought frequency, duration and intensity increase over time. Values of static and dynamic drought indices increase over time, with more frequent occurrences of maximum index values equal to 1, especially towards the end of the century (2071–2100). Spatially, the increase occurs over almost the entire area, except in the more humid northern Spain, and in areas that are already dry at present, which are located in southeast Spain and in the Ebro valley. This study confirms the potential of static and dynamic indices for monitoring and prediction of drought stress.

**Keywords:** climate change; drought stress; drought monitoring; plant water stress; Spain

## 1. Introduction

Climate change is one of the greatest future challenges for society as a whole, and for agricultural production and food security specifically [1]. If the current situation of greenhouse gas emissions continues, agricultural productivity will be significantly affected, with temperature increases and rainfall decreases offsetting benefits of increased carbon dioxide concentrations [2]. Arora [3] reports an estimated 20–45% decline in maize (*Z. mays*) yields, 5–50% in wheat (*Triticum* L.) and 20–30% in rice (*Oryza sativa*). In Europe, a series of research projects of the Joint Research Centre, called PESETA, have analyzed in more detail the climate change impacts on a wide range of environmental and socio-economic aspects [4], showing clearly very uneven impacts within the EU and an important north-south divide. Southern Europe is hit especially hard, with a significant decline in agricultural production, whereas some parts of eastern and northern Europe show production increases. However, this type of assessment of crop yield under climate change is entirely based on crop-growth models, which often do not allow to take into account climate extremes, such as drought or freezing events. Trnka et al. [5] evaluated future agroclimatic conditions in Europe and found worsening conditions in most of Europe, including eastern and northern Europe. They attribute this to a higher risk of extremely unfavorable years that are likely to increase everywhere, for example, due to for example drought stress, frost, or the absence of snow cover that does not protect against frost.

It is well known that extreme events, such as heat waves and droughts, are projected to increase over the next decades [1]. Lu et al. [6] evaluated the uncertainty related to agricultural drought predictions from the Coupled Model Intercomparison Project phase 5 (CMIP5) models at a global

scale. They identified Europe and the Mediterranean as being among the hotspots, where droughts are expected to increase most notably for all analyzed emission scenarios. Pausas and Millan [7] indicate that in the Mediterranean the situation might be more complex because, due to socio-economic feedback loops, land abandonment of less suitable agricultural areas might actually lead to vegetation recovery, or greening, as opposed to the drying or browning process caused by human-induced climate change.

Different studies have addressed the increase of droughts in Europe and in the Mediterranean region under projected climate change scenarios. In order to correctly assess the impact of future climate change on agricultural droughts, it is crucial to use adequate drought indicators and work at the finest spatial scale possible. Firstly, drought indicators need to take into account both the decrease in rainfall, as well as the increase in temperature and evapotranspiration. Especially for agricultural purposes, it is crucial to have a correct representation of buffering soil moisture dynamics. Many studies, for example, have assessed the occurrence of meteorological droughts under future climate change, using the Standardized Precipitation Index (SPI), which only accounts for changes in the rainfall regime. Secondly, climate change projections are normally based on simulations from general circulation models (GCMs) that are run under various emission scenarios. The results however cannot be directly applied to climate change impact studies, and further downscaling is needed [8]. Higher resolution can be obtained by regional climate models (RCMs), nested within a GCM, but these generally inherit the biases and other deficiencies of the large-scale model and further, statistical, downscaling is needed. The basic idea behind statistical downscaling is to define a relationship between the large-scale model (either GCM or RCM) and the local climate [8].

Several studies analyze meteorological drought by means of RCMs. For instance, Maule et al. [9] used the SPI and a version of the Palmer drought severity index (PDSI) to analyze drought representation by 14 RCMs from the ENSEMBLES project [10] at a European scale. They conclude that, at a European scale, the results seem robust but warn to use quantitative results at smaller, regional scales. RCMs are also used frequently in soil moisture and hydrological drought analyses. Spinoni et al. [11] used climate predictions from the EURO-CORDEX to evaluate drought events at a European scale, by means of standardized precipitation index (SPI), standardized precipitation evapotranspiration index (SPEI) and the reconnaissance drought indicator (RDI). More detailed studies were carried out in different countries, using the data from the EURO-CORDEX project. For example, Meresa et al. [12] studied hydro-meteorological drought in ten Polish catchments by computing SPI, the Standardized Precipitation Evapotranspiration Index (SPEI) and runoff standardized indices for 1971–2100. They concluded that SPI and SRI indicated wetter conditions in the future, while SPEI indicated a drying trend. Potopová et al. [13] also used results from eight RCMs from the EURO-CORDEX project to calculate future drought trends in the Czech Republic, by means SPI and SPEI. Barrella-Ortiz and Quintana-Seguí [14] evaluated drought representation and propagation in three RCMs from the Med-CORDEX database, focusing on the Mediterranean region. They conclude that RCMs are a suitable tool for meteorological drought studies, but that they should be used cautiously for soil moisture and hydrological drought analyses.

Drought studies using statistically downscaled climate projections are much rarer, and have not been done for Spain, according to the knowledge of the authors. In addition, recent studies have shown that for drought prediction and monitoring it is crucial to take into account the buffering effect that soil properties have, as well as crop type and cropping characteristics. Jiménez-Donaire et al. [15] recently analyzed the potential of two new indicators, static and dynamic drought stresses, based on earlier work by Porporato et al. [16]. In a study under cereal in Southern Spain, they concluded both indicators identified agricultural droughts well and were found to be good predictors of crop yield.

The objective of this study is to analyze the effect of climate change on agricultural drought in Spain, using these novel static and dynamic drought stress indicators and statistically downscaled climate change predictions for the period 2006 to 2100.

## 2. Materials and Methods

### 2.1. Study Area

This study was conducted throughout mainland Spain and the Balearic Islands. The time period was limited to 2006–2100, because of the availability of statistically downscaled regional climate model (RCMs) projections.

Spain can be subdivided in three main biogeographical regions (Figure 1). Most of the country is classified as Mediterranean, with the exception of the northern coastal region that is Atlantic and the Pyrenees mountain range, which is classified as Alpine [17]. The Mediterranean biogeographical region corresponds to a temperate climate region (type C), according to the Köppen classification system, and can be further subdivided into hot and warm summer Mediterrean Climate, Csa and Csb, respectively, which are considered typically Mediterranean climate zones, and cool-summer Mediterranean climate, Csc. Some parts of this biogeographical region of Spain are also drier and are also classified as dry climates (type B, specifically hot deserts climate, BWh, cold desert climate, BWk, and hot semi-arid climate, BSh). These are located in the southeast coastal regions of Murcia, Almeria and Valencia and the Ebro valley. The northern Atlantic biogeographical region mostly corresponds to temperate climate (type C) without a dry summer (humid subtropical and oceanic climate, Cfa and Cfb). The Alpine regions correspond to cold climate types without dry season (warm summer continental and subarctic climate, Dfb and Dfc).

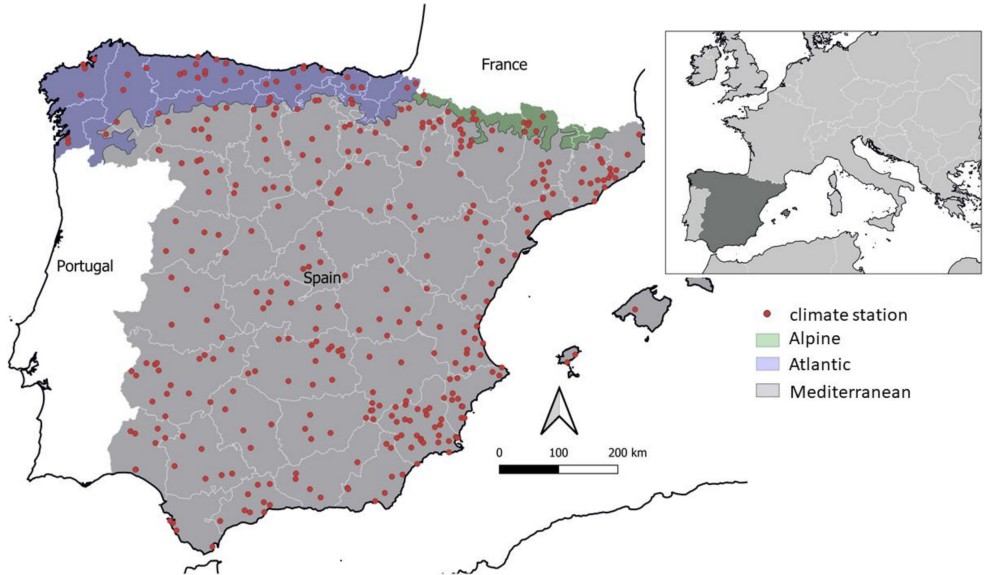

**Figure 1.** Biogeographical regions and location of used climate stations within Spain. Inset shows Spain within Europe. White lines indicate province boundaries.

### 2.2. Climate Change Data

In this study, statistically regionalized projections of climate change are used that were developed by the Spanish National Agency for Meteorology. Daily precipitation, maximum and minimum temperature data between 2006 and 2100 were used from the model HadGEM2-CC, under the scenario RCP8.5, from the Intergovernmental Panel on Climate Change's fifth assessment report. These data are available for free download [18]. This study only evaluates this emission scenario, as no statistically downscaled data for Spain were available at present for this model under other scenarios, such as, for example RCP4.5, which is another frequently used scenario that establishes a more moderate greenhouse gas increase. The scenario RCP8.5 used here assumes that emissions continue to rise at the present level throughout the 21st century. This scenario is generally taken as the basis for the worst-case climate change scenario and, while it has received some criticism by some reports

because it is considered implausible, it is useful, as it allows tracking and predicting the effect of our current behavior, and makes it possible to demonstrate the impact of emission reduction policies if no action were to be taken. In addition, some recent research has concluded that this scenario with a high temperature increase at the end of this century is becoming increasingly more plausible because of feedback effects [19–21]. In total, daily data between 2006–2100 was available for 374 stations. The distribution of these stations is shown in Figure 1.

Calculations of soil water balance and drought stress indicators were made for each of these points in R 4.0.0 [22] with packages dplyr, tidyverse and yarrr. Output maps were generated in QGIS 3.10.6 [23].

## 2.3. Drought Stress Indicators: Static and Dynamic Stress

A full description of the calculation of the static and dynamic stress drought indicators is given in Jiménez-Donaire et al. [15]. Static stress ($\zeta$) is proportional to the excursion of soil moisture ($W$) below a critical point that corresponds to incipient stomatal closure ($W^*$), and reaches a maximum equal to 1 for soil moisture values below permanent wilting point ($W_{pwp}$). It is calculated as:

$$\begin{cases} \zeta(t) = 0, \text{ for } W(t) > W^* \\ \zeta(t) = 1, \text{ for } W(t) < W_{pwp} \\ \zeta(t) = \left[ \frac{W^* - W(t)}{W^* - W_{pwp}} \right]^q, \text{ for } W_{pwp} \leq W(t) \leq W^* \end{cases} \tag{1}$$

Plant stress can increase non-linearly with soil moisture deficit, where the coefficient $q$ is a measure of this non-linearity. In this study a value of 1 was used, implying a linear soil moisture-stress relation. The static stress $\zeta(t)$ is calculated at daily time steps, and the overall static water stress, $\zeta$, is then calculated by integrating the individual positive values of $\zeta(t)$ over the whole year, excluding periods where $\zeta(t) = 0$.

Dynamic stress ($\theta$) includes information on the mean duration and frequency of drought periods. This indicator therefore extends the information contained in the static stress indicator, as the latter only takes into account the intensity of the droughts that occur over a year. The expression for $\theta$ is:

$$\begin{cases} \theta = \left( \frac{\zeta \, T_{W^*}}{k \, T_{seas}} \right)^{1/\sqrt{n_{W^*}}}, \text{ if } \zeta \, T_{W^*} < k \, T_{seas} \\ \theta = 1, \text{ otherwise} \end{cases} \tag{2}$$

where $n_{W^*}$ and $T_{W^*}$ are the number (-) and mean duration (days) of all drought occurrences over a year, respectively, $T_{seas}$ is the duration of the growing season (days) and $k$ is a parameter, set to 0.5 following Porporato et al. [16].

The soil water balance is calculated using the same approach as Jiménez-Donaire et al. [15], using a simple bucket model applied over the root zone, and taking into account the main processes of rainfall infiltration ($f$), evapotranspiration ($e$) and deep seepage ($g$). The calculated soil moisture values $W(t)$, are representative of the mean moisture content over the root zone depth, taken as 1m.

$$\frac{dW(t)}{dt} = f - e - g \tag{3}$$

The main difference with the previous study the future climate change prediction datasets are lacking detailed information on meteorological variables needed to calculate reference evapotranspiration using FAO Penman-Monteith's formula. Therefore, Hargreaves' formula [24,25] was used to calculate reference evapotranspiration from the minimum and maximum temperature data:

$$e_0 = AHC \, R_a (T + 17.8) \sqrt{(T_{max} - T_{\min})} \tag{4}$$

where $R_a$ is the water equivalent of extraterrestrial radiation (mm day$^{-1}$); $T$, $T_{max}$ and $T_{min}$ are the daily mean, maximum and minimum temperatures (C), respectively; *AHC* is the adjusted Hargreaves coefficient, equal to 0.0023 in the original equation. Although some studies have pointed to regional differences and have proposed locally calibrated values [26], this value was used for the entire study area as it is sufficiently robust [25] and no estimates are available for the different parts of Spain. Finally, the real daily evapotranspiration rate is calculated by correcting this potential rate, $e_0$, by the wetness of the soil profile, $\omega$, and the crop coefficient, $k_c$:

$$e = \omega k_c e_0 \tag{5}$$

Static and dynamic drought stress indicators were calculated for the period 2006–2100, and their evolution was analyzed over 3 time periods: 2006–2040, 2041–2070 and 2071–2100.

## 3. Results

### 3.1. Spatial and Temporal Trends in Static and Dynamic Drought Stress between 2006–2100

The behavior of two representative sites, respectively for the southern, semi-arid part of Spain, and for the northern, more humid part of Spain, is analyzed first to illustrate behavior of the drought indices and their evolution over the study period. Most of mainland Spain is characterized by relatively dry Mediterranean climate, with a clear dry season. Drought incidence is already a frequent phenomenon, as the observed year-to-year rainfall variability is relatively high. This means that years where rainfall falls below 500 mm are already frequent at present. This behavior reflects in the climate projections for the period 2006–2100 shown in Figure 2a, for the La Rambla site, Córdoba province, which can be taken as a representative site of this climate zone. The coefficient of variation of the annual precipitation is 31%. At the same time, the climate projections indicate a significant decrease in precipitation that amounts to 0.45% per year, translating in a decrease of the mean annual precipitation of 530 mm between 2006–2040 to 498 mm for 20341–2070, and decreasing even more strongly to 399 mm for 2071–2100. This is clearly reflected in the static and dynamic drought stress indices. While this semiarid zone has already an incidence of important drought events at the beginning of the study period, the occurrence of serious drought events, with values of static and dynamic stress hitting highs equal to the maximum value of 1, increases strongly by 2071–2100. The mean static stress increases from 0.49 in 2006–2040, to 0.57 in 2041–2070 and 0.76 in 2071–2100. Over the same periods, the dynamic stress increases from 0.35, to 0.44 and 0.69, respectively. The occurrences of dynamic stress values equal to 1, implying that soil moisture remains below the critical soil moisture level during the entire growing season, and increases from 2 to 4 to 18 times in the three different periods (2006–2040, 2041–2070 and 2071–2100).

In contrast, the stress indices in the northern Spanish site of Lugo, province of Lugo, shown in Figure 2b, are much lower due to a higher rainfall regime and the absence of a dry season. Even so, precipitation decreases significantly at a rate of 0.28% year$^{-1}$. Mean annual precipitation lowers from 997 mm between 2006–2040, to 952 mm between 2041–2070 to 831 mm for the last period 2071–2100. Overall drought stress levels are low, generally lower than the minimum values that are reached at the La Rambla site, so no crop stress should be expected. However, also at this site an increase can be observed over the studied period due to climate change, especially for the last period. The mean static stress index remains at 0.07 between 2006–2040 and 2041–2070, but rises to 0.18 for 2071–2100. The same behavior is seen for dynamic stress, almost stable at 0.03 and 0.04 between 2006–2040 and 2041–2070, respectively, and increasing to 0.12 for 2071–2100. However, static and dynamic stress never reaches 1, even in the years with minimum annual precipitation of the series.

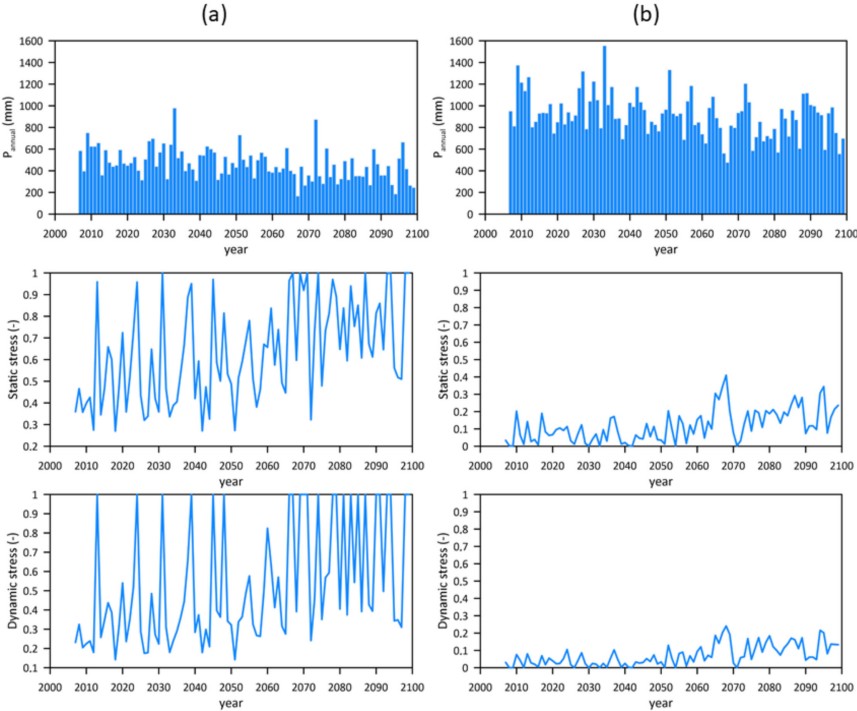

**Figure 2.** Evolution of precipitation, static stress and dynamic stress between 2006–2100, for a site in southern Spain and northern Spain, respectively: (**a**) site La Rambla, Córdoba (N 37.606, W −4.741); (**b**) Lugo, Lugo (N 43.011, W −7.555).

Spatial trends of static and dynamic stress for the periods 2006–2100 are shown in Figures 3 and 4. Static and dynamic stresses are classified in 5 levels: no stress, very low (0–0.2), low (0.25–0.50), moderate (0.50–0.75) and high stress (0.75–1.00). In the first period of 2006–2040, high levels of static and dynamic stress (0.75–1.00) are observed in the southeast of Spain and the Ebro valley, which are already characterized by a dry climate. The northern Atlantic region, with more abundant precipitation is characterized by no stress (no data) to very low stress values (0–0.25). Most of mainland Spain is characterized by low to moderate levels (0.25–0.50 and 0.50–0.75). Over the years, especially during the last period 2071–2100 it can be observed how most of mainland Spain evolves to increasing drought stress and more sites are characterized by high static and dynamic stress levels, over 0.75.

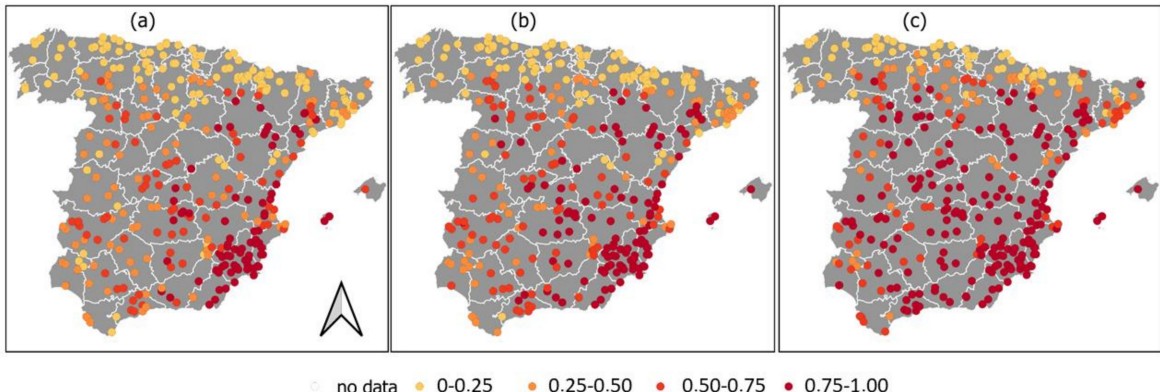

**Figure 3.** Average value of the static stress indicator in Spain over the periods 2006–2040 (**a**); 2041–2070 (**b**); and 2071–2100 (**c**).

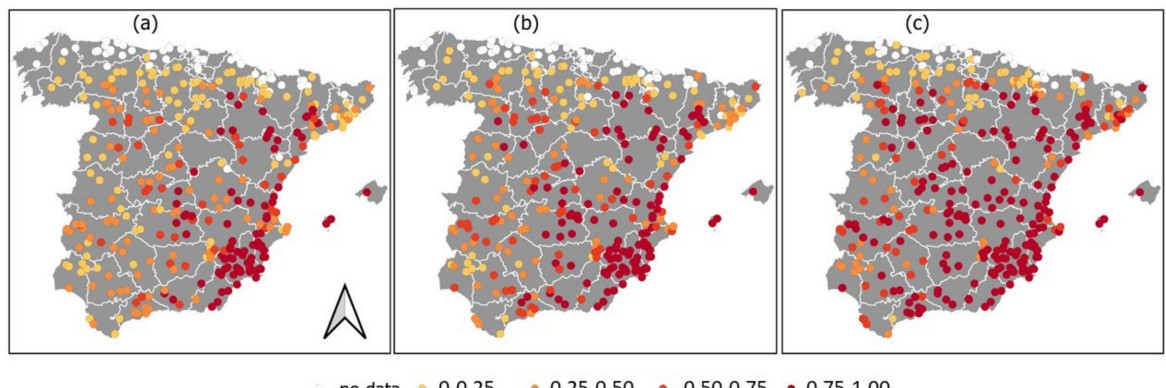

**Figure 4.** Average value of the dynamic stress indicator in Spain over the periods 2006–2040 (**a**); 2041–2070 (**b**); and 2071–2100 (**c**).

Figure 5 summarizes the temporal changes in static and dynamic drought stress, comparing the period 2006–2040 versus 2041–2070 (a and c), and 2006–2040 versus 2071–2100 (b and d). Both drought stress indicators show the same trends. Firstly, the sites in areas that are already dry, and characterized by high stress levels that, at present, show almost no change over time, or even a small decrease in mean drought stress, showing up in blue. These points are located along the southeastern Spain and in the Ebro valley. However, the rest of Spain shows a significant change with respect to the reference period 2006–2040. Change is moderate for the second period, 2041–2071, for most sites with increases contained below 30%. However, especially for the last periods, most sites show up in dark red, indicating changes of over 45%. For sites that start out with low absolute drought stress values, such as those in the north of Spain, this does not imply important problems with crop production, although some sites change from no stress condition to low stress condition. However, for most of mainland Spain, already characterized by moderate to high stress levels in 2006–2040, this increase indicates an alarming situation and flags problems with droughts affecting agricultural crop production in most of the country.

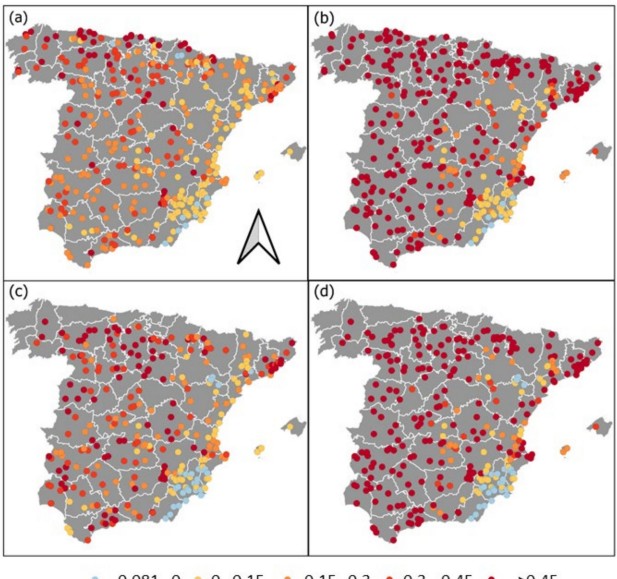

**Figure 5.** Increase of the static drought stress in Spain over the periods: (**a**) 2006–2040 vs. 2041–2070 and (**b**) 2006–2040 vs. and 2071–2100; and increase of the dynamic drought stress over the periods (**c**) 2006–2040 vs. 2041–2070 and (**d**) 2006–2040 vs. 2071–2100. Change is indicated as fractional and negative values indicate a drought stress decrease.

### 3.2. Analysis of Static and Dynamic Stress Index Dynamics

The relation between annual precipitation and static and dynamic stress is shown in Figure 6. It can be seen that the relation of both indicators with rainfall is highly non-linear and increases sharply below approximately 500 to 600 mm annual rainfall. Where rainfall is higher, it is very rare to have droughts. However, within this range of annual rainfall, the variation of static and dynamic stress indices is high, and can range from 0 (no stress) to 1 (maximum stress), depending on how the precipitation is distributed throughout the growing season. This illustrates clearly how rainfall alone is not a good indicator of agricultural drought stress.

The relation between static and dynamic stress is shown in Figure 7. It can be seen how most points are clustered, except for a few points that are limited by the maximum envelope, as, for static stress values higher than 0.8, dynamic stress reaches the maximum of 1.

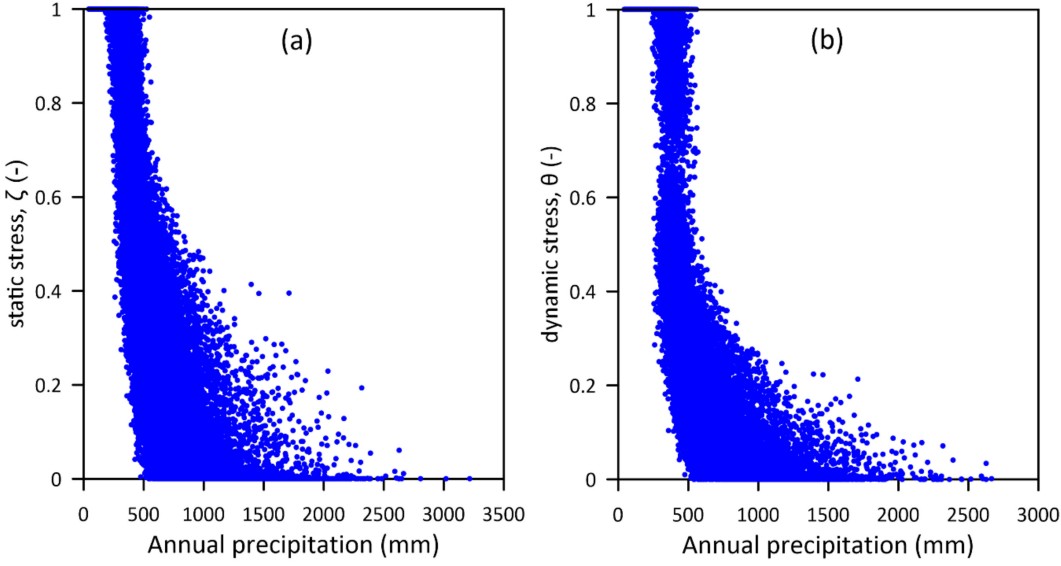

**Figure 6.** Relation between annual precipitation and (**a**) dynamic and (**b**) static stress for all sites and studied years

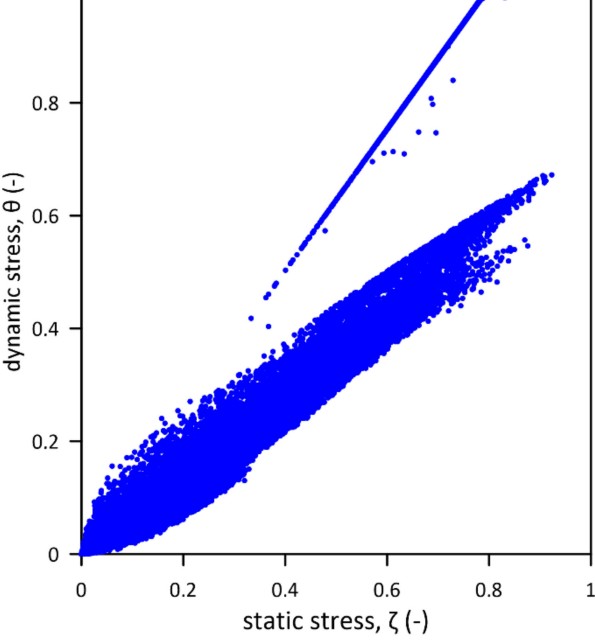

**Figure 7.** Relation between dynamic and static stress for all sites and studied years.

## 4. Discussion

In Spain and in the Mediterranean basin specifically, agriculture has always been affected by large natural climate variability and by droughts. Historically, Spain has suffered important droughts. Domínguez-Castro [27] analyzed Spanish drought episodes over the past 500 years, based on ceremonial records and tree rings. They reported frequent droughts since the start of their historical records in the 16th century, with the most severe droughts being recorded during the period from the end of the 16th century up until the 18th century. Additionally, in recent times, Spain has suffered several intense drought episodes [28], and some of the most notorious ones in terms of impact were produced during the periods: 1941–1945, 1979–1983, 1991–1995 and 2004–2007 [29]. Jiménez-Donaire et al. [30] studied drought incidence in the south of Spain between 2003 and 2013, and reported two severe drought periods (2004–2005 and 2011–2012) with associated crop damages between 70 and 95% of the agriculturally insured area. The concern of Spain with droughts is reflected by the presence of drought response measures and planning in policy tools, such as the National Hydrologic Plan Act [31].

This study demonstrates that the novel indicators static and dynamic drought stress, proposed by Jiménez-Donaire et al. [15], are useful for analyzing drought dynamics at regional level. The results, using statistically downscaled climate projections at regional level, indicate that if greenhouse gas emissions continue at the present level, drought occurrence will increase significantly between 2041–2070, and especially between 2071–2100, compared to the reference period of 2006–2040. In the Mediterranean areas, severe droughts with maximum values of static and dynamic droughts equal to 1 are shown to increase in magnitude, duration and frequency. Drought incidence increases over the whole country, except in the north and southeast. In northern, more humid regions, static and dynamic drought indices are below 0.25, and not limiting for crop growth, although they also experience an increase towards the end of the studied period. In the southeast and part of the Ebro valley, drought occurrence is already very high in the reference period 2006–2040.

Similar results were obtained by Spinoni et al. [32] using SPI and SPEI indices at European level. They considered three periods, 1981–2010, 2041–2070 and 2071–2100. For this last period, their results indicated more frequent and severe extreme droughts over the whole European continent, except Iceland, under the most severe emission scenario (RCP8.5) that was used here. They reported especially severe increases in southern Europe. For the Iberian Peninsula, a strong increase, meaning an increase of more than one additional event every 10 years, in more than 80% of the area for all seasons except winter, where a more moderate increase was observed. Our results are specifically designed to consider specifically the growing season only, with precipitation during fall, winter and spring being particularly important, whereas an increase of droughts during the summer dry period can be expected to have little or no effect on the evolution of soil moisture during the growing season, and therefore will also have little effect on static and dynamic drought indices. Marcos-Garcia et al. [33] studied climate change impact on meteorological drought and hydrological drought in the Mediterranean basin of the Jucar river. Although their predictions are geared towards the smaller river basin scale and at the mid-term (i.e., up to 2069), their results also show a similar trend using a normalized SPI and SPEI. They report a future decrease in the number of dry spells for the RCP8.5 scenario, but an increase of the average duration and intensity, meaning that often a single dry spell covers the entire analysis period instead of several, shorter dry spells. They also conclude that temperature effects on increase of evapotranspiration should not be ignored, and therefore SPEI is more useful than SPI. In our study, this effect was taken into account by resolving the full soil moisture balance over the period 2006–2100 for all studied sites. Gaitán et al. [34] studied future droughts in the Aragon region in northeast Spain, covering part of the Ebro valley, for two RCPs, RCP4.5 and 8.5. Their results, also based on the use of the SPI and SPEI indices, confirm the clear trend toward increasingly intense periods of droughts, especially towards the end of century for the period 2071–2100. Interestingly, they report this is only detected when considering SPEI, which in addition to precipitation takes into account evapotranspiration, but is softened in the SPI scenarios. At the spatial scale, they also observed the most affected region to be the Ebro valley.

Global change has been reported to lead to drought increase in several regions around the globe, In some areas, it is caused by a combination of temperature increase and precipitation decrease, as this study has shown for the case of Spain, while in other regions droughts increase, in spite of precipitation increase, due to the dominating effect of temperature increase. Wang et al. [35] studied the increase of drought frequency and characteristics in the Huai river basin in China. They used the SPEI index and found that although climate change models project an increase in precipitation, it was not enough to offset the increased evapotranspiration due to temperature increases. Similar to this study, they reported a slight increase in droughts at the beginning of the 21st century, and a strong increase towards the end of the 21st century. In other areas droughts might increase due to even more complex situations. In Poland in central Europe, Sojka et al. [36] report an increase in the extension of rain-free periods, in spite of an overall precipitation increase. Their study shows that this results in a decrease of mean groundwater levels, and a reduction of subsurface flow. They report in contrast an increase of extreme events, leading to more runoff, but this water cannot be stored in the soil and used for agricultural crop production. Amnuaylojaroen and Chanvichit [37] analyzed the tendency of agricultural drought under climate change for Mainland Southeast Asia, using the SPI and crop water need (CWN) indices. The compared present-day with the period 2020–2029 under the scenario RCP8.5. Again, their climate predictions favor drought increase for this region, due to the combination of precipitation decrease and temperature increase. However, they only reported a change in SPI, while their index CWN, which would, in theory, be better suited, as it takes into account evapotranspiration, did not indicate drought increase.

This study aimed to characterize drought patterns across Spain under the worst-case emission scenario RCP8.5. The use of RCP8.5 impacts our results, as it the high end of $CO_2$ emissions and temperature increase. While different authors consider such high temperature increases more and more likely, as discussed earlier [19–21], it would be useful to examine other, more conservative scenarios in future studies. Follow-up studies should also explore different climate models from the CMIP5 ensemble. In any case, our results show that static and dynamic drought stress indicators are very useful to evaluate drought stress under climate change. Static drought stress indicates the drought magnitude, and dynamic drought stress incorporates additional information on duration and frequency of these drought events. This latter indicator therefore allows to obtain information on different aspects of drought (magnitude, duration and frequency) with a single indicator.

## 5. Conclusions

The objective of this study was to evaluate the suitability of static and dynamic stress as indicators of agricultural drought stress, and to use these indicators to evaluate spatial and temporal patterns of agricultural drought stress under climate change in Spain. The results show that static and dynamic drought stress are highly suitable indicators. Static drought stress indicates the magnitude of drought stress, while dynamic drought stress also includes frequency and duration of drought events. Both are shown to increase in the 21st century, especially towards the end of the studied period. Changes are significant for most of mainland Spain, which is under a Mediterranean climate. Only in the southeastern areas that are already very dry and in the northern areas that are humid, is the impact of climate change on droughts absent to low.

The projected climate scenarios and the methodology used in this study have several limitations. For example, it is expected that, for crops and pastures, production will be delayed by the onset of autumn rainfall. Water scarcity and other climate-induced changes to the cropping cycle, such as for example phenological changes or chilling requirements, might also change the suitability of entire regions for certain crops altogether and force shifting cropping patterns. Such changes in the growing season would be very interesting to address in future studies. To include these adaptations of crop production, it is necessary to couple crop simulation models with socio-economic modelling. Such efforts are underway in the framework of large international modelling efforts, such as AgMIP8 [38] or MACSUR [39], but need to take into account extreme events such as the droughts modelled here.

**Author Contributions:** Conceptualization, T.V.; methodology, M.d.P.J.-D.; software, M.d.P.J.-D. and T.V.; formal analysis, M.d.P.J.-D.; investigation, M.d.P.J.-D. and T.V.; writing—original draft preparation, M.d.P.J.-D.; writing—review and editing, M.d.P.J.-D., J.V.G. and T.V.; visualization, M.d.P.J.-D. and T.V.; supervision, J.V.G. and T.V.; project administration, T.V.; funding acquisition, T.V. All authors have read and agreed to the published version of the manuscript.

**Funding:** This study was funded by the research projects AGL2015-65036-C3-2-R and PID2019-109924RB-I00 financed by the "Programas estatales de generación de conocimiento y fortalecimiento científico y tecnológico del sistema de I + D + i y de I + D + I orientada a los retos de la sociedad".

**Acknowledgments:** The climate data projections were obtained from the Climatic Services of the Spanish National Agency for Meteorology (Agencia Estatal de Meteorología), Ministry of Ecological Transition and Demographic Challenges (MITECO).

**Conflicts of Interest:** The authors declare no conflict of interest.

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
