# Peer review of "Impact of Climate Change on Agricultural Droughts in Spain"

_water, doi:10.3390/w12113214_

Round 1

Reviewer 1 Report

General comments:

In my opinion this paper is an interesting study and authors have collected a big dataset to analyze impact of climate change on Spain agricultural areas. The paper is generally well written and structured. The introduction is relevant, and theory based including systematic contribution to the research literature in this area of investigation. However, in my opinion the paper has some shortcomings/ disadvantages in regard to methodology and text, which I specified below.

Specific comments:

Abstract. Please add information that Regional climate model (RCMs) projections, including RCP8.5 scenario, were used in this study. It is necessary to clarify content of this paper.

Materials and methods (2.2. Climate Change Data). Why authors have decided to use only RCP8.5 scenario? In my opinion this is the greatness shortcoming in this study. RCP8.5 includes the most negative scenario for future climate changes but usually in studies concerning this research field also RCP4.5 is used. According to my knowledge RCP4.5 represent as an intermediate scenario so also should be analyzed. If authors decide to analyze only RCP8.5 please add why intermediate scenario was not applied in this study.

Materials and methods. Line 114. Please review article in terms of editing, for example authors should add the “.” between stations and The distribution.

Materials and methods. Figure 1. General location of Spain in the Europe should be added (as a small, additional map). Scale, north arrow and legend is missing. Also coordinated should be modified into UTM format.

Results. In some places description “Error! Reference source not found” is visible in the text (for example line 166 or 185), please correct it.

Results. Figure 2. Authors wrote “Spain can be subdivided in three main biogeographical regions.” (line 94). In my opinion in the figure should be presented graphs for three stations representative each of the three main biogeographical rerions.

Discussion. When introduction part is sufficient and describes good background of the problem, discussion should be improved. Authors should discuss obtained results on the background of scientific literature. What is the main contribution of current work to the scientific literature? Results should be compared with other similar studies in terms of methods used and application, for example:

  • Sojka, M., KozÅ‚owski, M., KÄ™sicka, B., WróżyÅ„ski, R., Stasik, R., NapieraÅ‚a, M., ... & Liberacki, D. (2020). The Effect of Climate Change on Controlled Drainage Effectiveness in the Context of Groundwater Dynamics, Surface, and Drainage Outflows. Central-Western Poland Case Study. Agronomy, 10(5), 625.
  • Wang, J., Lin, H., Huang, J., Jiang, C., Xie, Y., & Zhou, M. (2019). Variations of Drought Tendency, Frequency, and Characteristics and Their Responses to Climate Change under CMIP5 RCP Scenarios in Huai River Basin, China. Water, 11(10), 2174.
  • Amnuaylojaroen, T., & Chanvichit, P. (2019). Projection of near-future climate change and agricultural drought in Mainland Southeast Asia under RCP8. 5. Climatic Change, 155(2), 175-193.

Conclusion part is missing and must be added. In my opinion, last paragraph of discussion (starting from “The projected climate scenarios…”) should be moved into conclusion part.

Reviewer 2 Report

I have evaluated the manuscript water-992366 entitled “Impact of Climate Change on Agricultural Droughts in Spain” submitted for potential publication in Water. The idea of study is interesting and has worth of publishing but manuscript has some serious shortfalls which need careful revisions to make it in acceptable form. Main concerns are:

  1. The authors should expand the scope of study rather than only focusing on Spain. They must highlight area of such type of climate where these findings are implicated as well. In that way the manuscript will attract more citations at international level. In its current form it better fits in some local Spanish journal.
  2. Novelty of study must be elaborated in more details as many studies with similar type of objectives are already available.
  3. The authors should correlate this climate change induced agricultural drought on crops productivity and management options to tackle it; this will be the important findings of this study.
  4. The authors should do some statistical or meta-analysis to make results more interesting.
  5. Mechanistic elaboration of results is missing in discussion section. Currently discussion section is just like repetition of results along with its mere comparison with earlier studies.
  6. Add concrete conclusion section
  7. Add scientific name of each crop at its 1st For example three crops are enlisted in line 32 without their scientific names; check whole section.
  8. Write in full once before using abbreviations. For instance at line 33 see PESETA; check whole manuscript.

Reviewer 3 Report

Review of the submitted manuscript entitled 'Impact of Climate Change on Agricultural Droughts in Spain'.

The content of the manuscript concerns one of the biggest contemporary global problems, which is the increasingly noticeable shortage of water and agricultural droughts resulting mainly from global warming (but not only). The aim of the research was' evaluation the impact of climatic changes on agricultural droughts, using static and dynamic drought indices. The study used meteorological data from 374 meteorological stations in Spain. In line with what was to be expected, research has shown that the frequency, duration and intensity of droughts will increase in this century. The research also indicated the usefulness of static and dynamic indicators in drought monitoring.

In the Introduction, the research problem is well presented and the need for such research is justified. However, the information that in central Europe may be expected to increase in agricultural production must be corrected. Please note that for several years this region has experienced severe droughts during the growing season, which significantly reduce the crop yields. In addition, the lack of snow cover in winter causes winter plants to freeze and further losses are caused by late spring frost (i.e. maize crops). Moreover, please note that in central Europe over the past centuries there have been much more severe droughts than the ones currently observed, i.e.

Przybylak R, OliÅ„ski P, Koprowski M, et al (2020) Droughts in the area of Poland in recent centuries in the light of multi-proxy data. Clim Past 16:627–661. doi: 10.5194/cp-2019-64 (see table 3, 4 and 5)

Brázdil, R., Dobrovolný, P., Luterbacher, J., Moberg, A., Pfister, C., Wheeler, D., & Zorita, E. (2010). European climate of the past 500 years: new challenges for historical climatology. Climatic Change101(1-2), 7-40.

It is possible that soon, as a result of climate change (but also hydrological changes), we can expect equally drastic and long-lasting droughts in central Europe.

L.105-109 There is no justification for the selection of GCMs. A lot of GCMs have been published and they are very different.

L.108 These data are available for free download. - This sentence is unnecessary.

L.119 The numbers of R and QGIS software versions should be given. Also, if any additional R packages are used, they should also be quoted.

L.166 Error! Reference source not found

L.185 Error! Reference source not found

L.242 The authors show the results of the relationship between stress and annual rainfall. Wouldn't it be more accurate to analyze stress in relation to the growing season only?

The Discussion is generally well written. However, the reader may get the impression that the drought theme is a fairly recent problem in Spain and that droughts have only emerged in recent decades. This is certainly not true. The only thing that is constant about the climate is that it keeps changing. It should be mentioned here.

Kind regards

Round 2

Reviewer 1 Report

The manuscript has been improved following the recommendations of the Reviewer; most of my concerns have been addressed and justified.

In my opinion this paper is an interesting study and authors have collected a big dataset to analyze impact of climate change on Spain agricultural areas. The paper is generally well written and structured.

I believe that manuscript in present form can be accepted to publish in Water Journal.

Reviewer 2 Report

The authors have incorporated the comments and now I have no more comments. 

Reviewer 3 Report

I'm fully satisfied in Author's responses and new version of manuscript. In my opinion the manuscript is suitable for publication in Water journal.